# Woven organic crystals

Linfeng Lan[1], Liang Li [2,3], Jianqun Qi[1], Xiuhong Pan[1], Qi Di[1], Panče Naumov [2,4,5,6] ✉ & Hongyu Zhang [1] ✉

Woven architectures are prepared by physical entanglement of fibrous components to expand one-dimensional material into two-dimensional sheets with enhanced strength and resilience to wear. Here, we capitalize on the elastic properties of long organic crystals with a high aspect ratio to prepare an array of centimeter-size woven network structures. While being robust to mechanical impact, the woven patches are also elastic due to effective stress dissipation by the elasticity of the individual warp and weft crystals. The thermal stability of component crystals translates into favorable thermoelastic properties of the porous woven structures, where the network remains elastic over a range of 300 K. By providing means for physical entanglement of organic crystals, the weaving circumvents the natural limitation of the small size of slender organic crystals that is determined by their natural growth, thereby expanding the prospects for applications of organic crystals from one-dimensional entities to expandable, two-dimensional robust structures.

Dating back over seven millennia[1], weaving is one of the oldest crafts known to humankind. The earliest evidence of woven objects comes from the Paleolithic era, when humans used roots and plant epidermis to weave webs for carrying stones that were used for hunting. The creation of woven structures such as fabric or cloth has a two-fold purpose: practical and ornamental[2,3]. The traditional weaving techniques, many of which have remained in common use till the present day, are based on wrapping, winding, knotting, knitting, crocheting, or braiding of yarns made of plant fibers or animal hair[4], and are mainly used to make domestic or technical textiles for thermal insulation and protection[5]. The extraordinary mechanical properties and visual appeal of some of these common wearables have recently inspired the development of strategies for weaving at a molecular scale[6–8]. The process of weaving requires a yarn made of either natural or synthetic (polymer) fibers that are reasonably elastic and can be bent without fracture while it is entangled into a two-dimensional structure[9]. With the projected shift from stiff to flexible devices, woven optoelectronic textiles have been identified as a prospective platform for wearable electronics, with increasing demand for e-textiles that are light in weight, flexible, and comfortable to wear[10,11]. The feasibility of such elements has been demonstrated in applications that include wearable supercapacitors[12], sensors[13], storage devices[14], light-emitting diodes[15], and transistors[16]. Although some of these applications require organic materials such as powders or glasses, they have not yet made use of organic crystal materials that carry an overlooked potential rooted in the lack of major structural defects, while they come with the benefits of having periodically ordered structures, optical transparency, and structure-properties anisotropy[17]. Recently, it has been realized that weak intermolecular interactions between discrete molecules render slender molecular crystals remarkably flexible[18–20] and dynamic[21–28] — properties that are now being explored for flexible optoelectronic devices such as active and passive waveguides[29–33], lasers[34,35], sensing technology[36,37], reconfigurable optical circuits[38,39], and other applications that require smart materials[40–42]. One of the major limitations of molecular crystals, however, is that, unlike polymers and inorganic glasses that can be pulled into fibers when soft, they can only be grown to small sizes that are limited, among other factors, by the crystallization kinetics. As such, they have not been considered competitive materials to polymers, elastomers, and resins, which in principle can be morphed into any arbitrary shape and size. Capitalizing on the

[1]State Key Laboratory of Supramolecular Structure and Materials, College of Chemistry, Jilin University, 130012 Changchun, People's Republic of China. [2]Smart Materials Lab, New York University Abu Dhabi, PO Box 129188 Abu Dhabi, UAE. [3]Department of Sciences and Engineering, Sorbonne University Abu Dhabi, PO Box 38044 Abu Dhabi, UAE. [4]Center for Smart Engineering Materials, New York University Abu Dhabi, PO Box 129188 Abu Dhabi, UAE. [5]Research Center for Environment and Materials, Macedonian Academy of Sciences and Arts, Bul. Krste Misirkov 2, MK–1000, Skopje, Macedonia. [6]Molecular Design Institute, Department of Chemistry, New York University, 100 Washington Square East, New York, NY 10003, USA. ✉e-mail: pance.naumov@nyu.edu; hongyuzhang@jlu.edu.cn

extraordinary elasticity of some organic crystals, here we apply one of humankind's oldest techniques, weaving, to prepare expanded two-dimensional structures that conceptually resemble fabrics. We also explore the correlation between the size of individual component crystals and properties such as width and length, tightness, coverage factor, and density of the ensuing crystalline woven structures. Further, we demonstrate that a variety of weaving methods[43] can be applied to crystals, including plain, twill, and satin weave having warp and weft yarns of different densities. The optical transparency of individual crystals is conserved, and the crystalline patches are transmissive to light and can be used to perform logical operations by selective excitation of different interlacing points. Inspired by some other examples of elastic crystals that have been recently utilized for electronic applications[44,45], this approach is promising for the fabrication of composite woven crystal networks with conductive capabilities that are expected and designed to function as logic circuits.

## Results and discussion

A woven fabric consists of two sets of parallel yarns that are interlaced with each other, of which one acts as a warp and the other as a weft[43]. The choice of the chemical composition of crystals was set to four chemically unrelated organic compounds **A–D** in Fig. 1a, which are all known to readily produce elastic and optically transmissive single crystals that are also conveniently fluorescent for enhanced visualization[26,46–48] (Fig. 1b and Supplementary Figs. 1–5). Crystals of **D** were manually twisted to explore incorporation of crystals with a non-conventional habit. The fluorescence quantum yields of the four types of crystals were measured for both short (less than 1 mm) and long (about 1 cm) crystals (Supplementary Table 1). The results showed that the quantum yields of the longer crystals were slightly reduced compared to those of the short crystals, an observation that underscores the minor impact of defects present in the longer crystals on their overall luminescent properties. The electronic spectra of the crystals provided the optimal conditions for light transmission and excitation (Supplementary Fig. 6). In the course of this work, thousands of long needle-like crystals were obtained by using controllable crystal growth methods, which were then screened for their morphology and optical transmission properties. From these, a certain number of crystals that can be used as optical waveguides were selected (Supplementary Fig. 7). These crystals are relatively uniform in size and have a smooth surface, which suggests high quality with a small number of defects (Supplementary Figs. 7 and 8). In a future application, the process of selecting crystals of sufficient quality can be automated and therefore more efficient. To prepare support analogous to a weft, organic single crystals of similar length (2–5 cm) were equidistantly aligned parallel to each other and affixed by gluing one of their ends to a base (Fig. 1c). Another set of crystals of similar size were then interlaced similar to a warp, producing an entangled structure with a topology analogous to that of a plain weave (Fig. 1c). The glue was then removed from the weft of the porous structure by treatment with boiling water. The weave density was increased and the net was tightened by pushing the warp and weft crystals (which are insoluble in water) closer to each other. To assess the potential detrimental effects of this treatment on crystallinity, single crystal X-ray diffraction analysis was performed on a crystal of **C** before and after immersion in boiling water. The diffraction data did not show any detectable deterioration by the treatment with boiling water (Supplementary Table 2). The interlaced structure was finally fixed by gluing the intersection of the outermost warp and weft crystals to prevent unweaving.

The resulting crystalline woven structure is a plain-weave-like entangled network, with each meridional crystal passing alternately below and above each zonal crystal. As shown in Fig. 1d, a crystalline patch **A** with five crystals in both the warp and the weft (5 × 5) prepared by this method is mechanically robust and can be hung without falling apart. The spacing between the parallel crystals is not strictly uniform,

as expected from their slightly different thicknesses (Fig. 1e; Supplementary Fig. 9). The woven patch can be expanded by interlacing new crystals, and the length of the patch is only determined by the length of the crystals. For instance, Fig. 1g shows a larger (20 × 18) patch of woven **A** of size 3.5 × 2.7 cm. The excellent elasticity of the individual crystals is retained within the crystalline patch (Fig. 1f), where their elasticity accounts for the weave's compliance with the slight bending required to straighten up the network (Fig. 1h). If kept within the elastic limit of the crystals, the crystalline patch can be bent or even curled manually an indefinite number of times and recovers its planar shape repeatedly after the force has been retracted without visible damage to the individual crystals (Fig. 1i, j).

The availability of crystals with elongated habit via the established crystal growth techniques poses a notable challenge to the selection of crystals for weaving, where the experimenter usually has limited control. However, the proposed weaving method is not restricted by the chemical composition, width, and thickness (generally within a few hundred microns) of the crystals, as long as they are slender, long (centimeter size) and reasonably elastic. The mechanical strength of a regular fabric depends on a variety of factors, including the properties of the yarn fibers, its thickness and elasticity, the method used for interlacing, and the density of the yarns, among others. To examine the effect of crystal dimensions, eleven 5 × 5 or 6 × 6 woven crystalline patches were prepared by using crystals **A–D** or their combinations: **A**1, **A**2, **B**, **C**1, **C**2, **C**3, **D**, **A**#**B**, **A**#**C**, **B**#**C**, and **A**#**B**#**C** (the ordinal numbers here refer to different samples from the same compound, while the hashtag sign refers to hybrid patches made of crystals of different compounds, Fig. 2a–d; Supplementary Figs. 9 and 10). Furthermore, the crystals in the patch of **D** were twisted along their longest crystal axis before weaving. The length of the helical pitch of all crystals of **D** ranged from 1.06 to 1.51 mm. The widths of the crystals were in the range 35–360 μm and their thickness was 9–95 μm (Supplementary Tables 3 and 4).

The average width and standard deviation of the crystals in the patches were assessed and compared to the dimensions of reported fibrous materials that are used for weaving (Supplementary Table 5). The basic characteristics of the woven structures are summarized in Supplementary Figs. 9–12 and Supplementary Table 6. The minimum grid size was 0.41 × 0.52 mm and the maximum was 1.78 × 1.83 mm. The width and length of the patches were between 2.12 and 8.70 mm. The density of the crystalline yarn was 57–236 crystals per 10 cm. The patch tightness was calculated to be 5.47–25.73%, the coverage factor was 10.58–42.35%, and the bulk density was 0.25–2.50 g cm$^{-3}$. Some of these parameters that relate to the woven structures are clearly correlated with some of the dimensions of the crystals. The relationship between the average grid (void part) or average pixel (void and surrounding crystals part) areas and the average densities of crystalline yarns (line density) and bulk density were also analyzed (Fig. 2e, f). The linear density and bulk density tend to decrease with the increase of the grid or pixel area.

The type of weave pattern in common textiles can be plain, twill, or satin (Fig. 3a, d, g)[1]. All of the crystalline patches described above are plain weave, which is also the most common pattern found with artificial composite materials. The plain weave is also known to be the simplest. It is reproducible at larger sizes and can be obtained and identified easily (Fig. 3b, c). By using the same method with 10 crystals of **A** and 10 crystals of **C** of known width and thickness, crystalline patches with different weave patterns were prepared (Fig. 3b, c, e, f, h, i; Supplementary Table 7). The twill weave is comparatively more stable than the plain pattern, and has a lower propensity to curl, finer surface, and slightly improved mechanical properties; however, it has a looser structure (Fig. 3e, f). The satin weave is essentially a twill weave that has been adjusted to reduce the intersection of the weft and warp to create a tightly woven fabric (Fig. 3h, i). The six characteristic parameters mentioned above were also evaluated to compare the

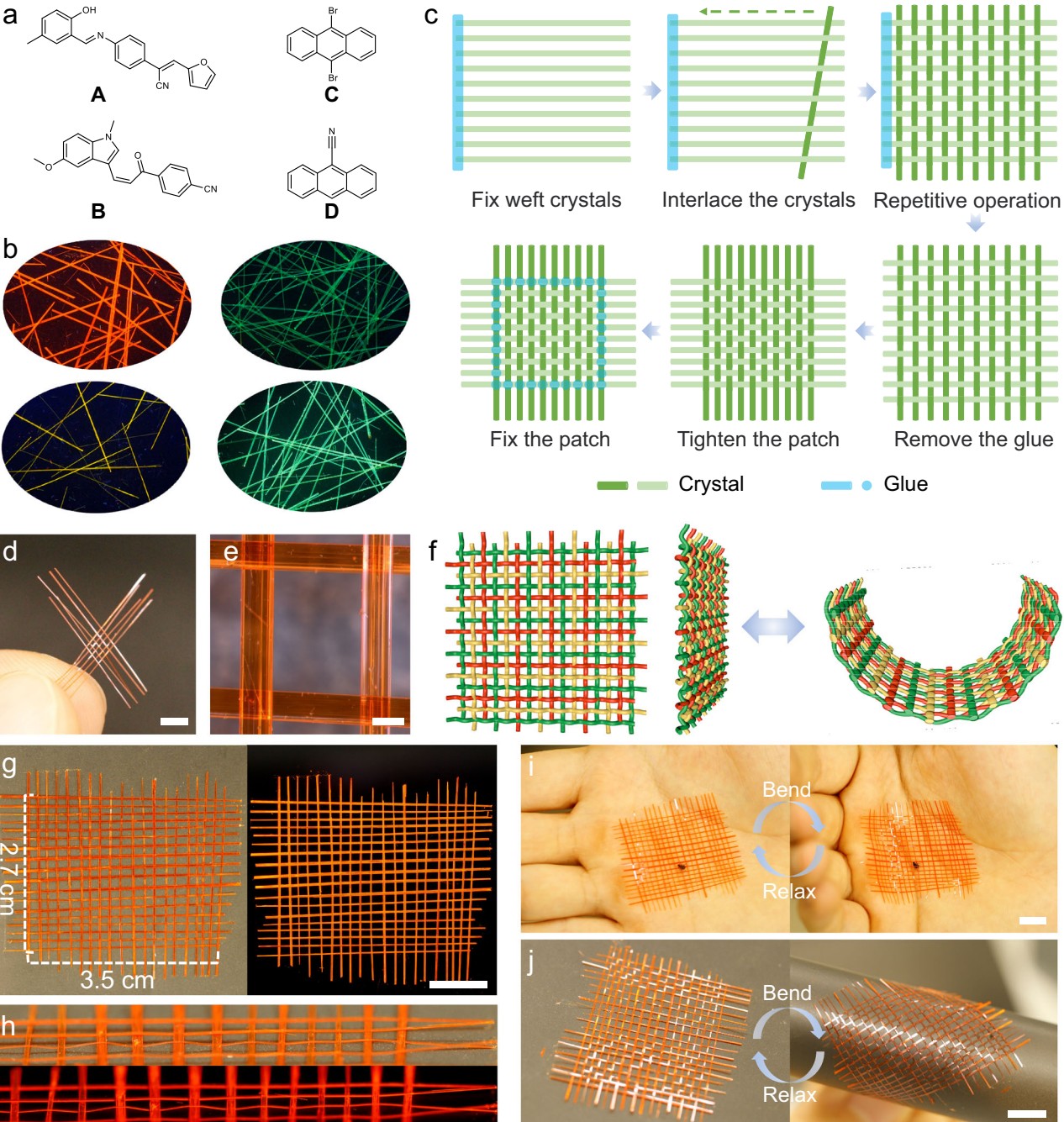

**Fig. 1 | Weaving organic crystals. a** Chemical structures of compounds **A**–**D** used to weave the crystalline patches. **b** Photographs of crystals of **A**–**D** recorded under UV light for contrast against a black background. **c** The method used to weave the crystals. The weft, which is composed of quasiparallel crystals, is set first, followed by interlacing crystals normal to the first set that act as a warp. The weft is then released from the base, the patch is tightened, and the outermost crystals are affixed with glue at the interlacing points to prevent the patch from disassembling. **d**, **e** Photographs of a crystalline patch of **A** (5 × 5) held between fingers (**d**) and a zoomed image of one cell with a hole in its grid (**e**). The scale lengths are 2 mm in panel **d** and 200 μm in panel **e**. **f** Schematic showing the curling of the crystalline patches. **g** A larger woven crystalline patch of **A** (20 × 18) in daylight (left) and under UV light (right). **h** Side view of the crystalline patch in daylight (top) and under UV light (bottom). The scale lengths in panels **g**, **i**, and **j** are 1 cm. **i**, **j** Bending and unbending of the patch of woven **A** supported by the palm (**i**) or by a piece of black paper (**j**).

effects of different weave patterns (Fig. 3j–n; Supplementary Table 8). Going from plain to twill to satin structure, the grid size and patch size decrease, while the line density, patch tightness, coverage factor, and bulk density increase. The warp crystals can be made denser and the weft crystals can be made sparser to obtain a tighter fabric (Fig. 3o). Plainly woven crystalline patches **C**4 (27 × 5) and **C**5 (47 × 4) were also prepared, and their parameters were also evaluated (Fig. 3p, q; Supplementary Table 9). Compared to the patches **C**1–**C**3 described above, **C**4 and **C**5 have a significantly tighter structure (Supplementary

Fig. 13). By sacrificing some weft density (only 24 and 17) and tightness (only 3.77% and 13.06%), the coverage of such patches were increased from 14–40% to 73–75%, and their bulk density increased from 0.59–1.80 g cm⁻³ to 10.81–32.25 g cm⁻³.

The process of weaving is well known to increase the mechanical strength of the material as well as its energy absorption capacity[43]. Slipping of the yarn is the main cause of both deformation and failure in regular fabric. This process of sliding the yarns that make up the fabric relative to each other causes drastic changes in both the

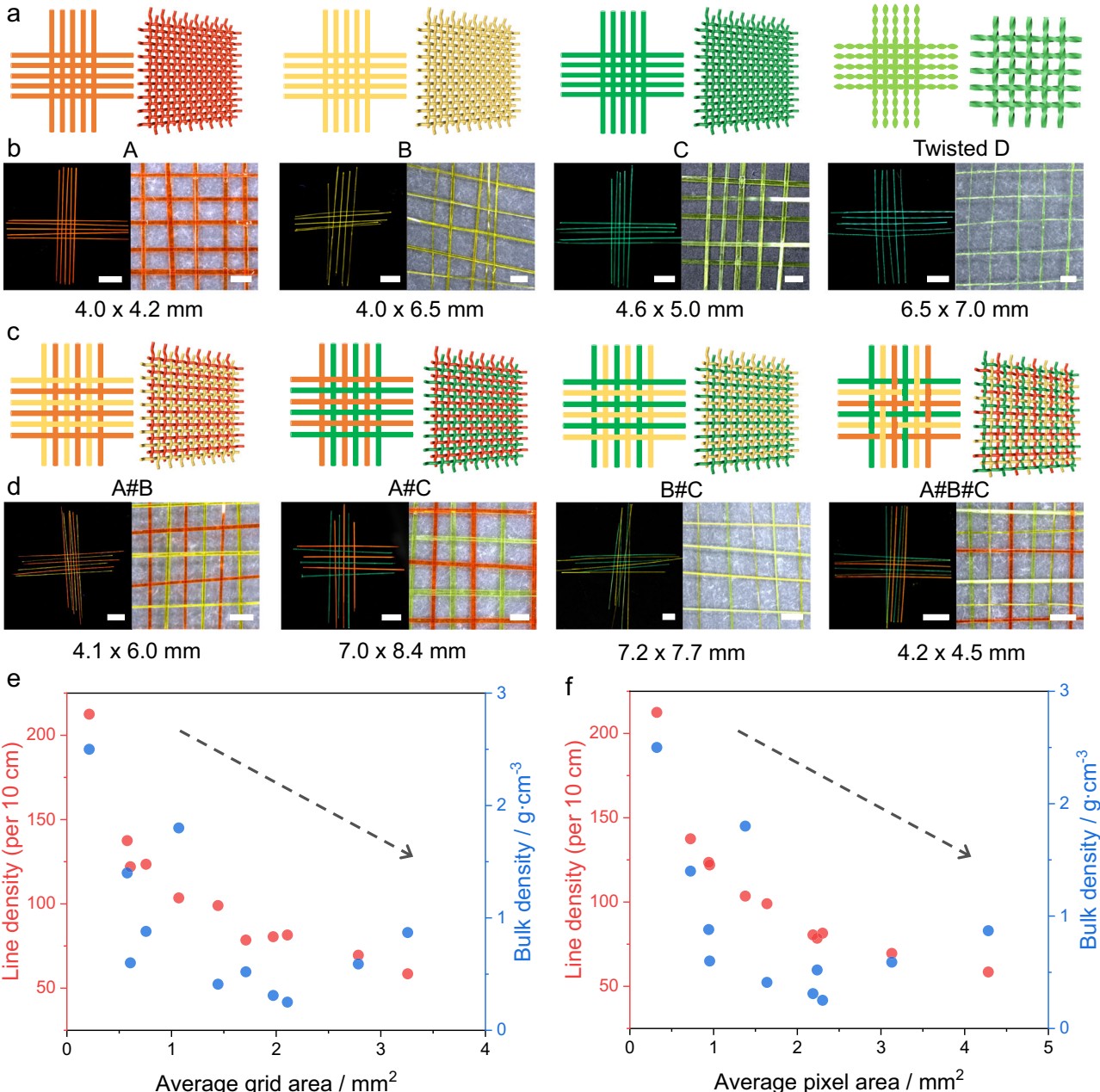

**Fig. 2 | Variety of woven crystals and basic characterization of the crystalline patches. a, c** Two (left) and three (right) dimensional schematics of crystalline patches (5 × 5 or 6 × 6) including those made of crystals **A**, **B**, **C**, **D** (twisted), **A#B**, **A#C**, **B#C**, and **A#B#C**. The orange red, yellow, green, and cyan lines represent crystals of **A**, **B**, **C**, and **D**, respectively. **b, d** Fluorescence (**b**) and magnified optical (**d**) photographs of patches **A**, **B**, **C**, **D** (twisted), **A#B**, **A#C**, **B#C**, and **A#B#C**. The scale length in panels **b** and **d** is 5 mm under daily light (left) and 1 mm under 365 UV light (right). **e** Scatter plots of average grid areas (*x*-axis) and line density and bulk density (*y*-axis). **f** Scatter plots of average pixel areas (*x*-axis) and line density and bulk density (*y*-axis). The black dotted line shows the trend.

mechanism of absorption of mechanical energy by the fabric and the yarn density[43]. As shown in Fig. 4a–c, the plain woven crystalline patch was stabilized against disassembly by using one of two approaches−by either fixing it at the four corners or by fixing it at the interlacing points along its perimeter. The firmness of such a fixed patch and an unfixed sample was tested by violently swinging them back and forth in the air (Supplementary Movie 1). When the crystals are not fixed, they easily come off the weave, and the structure falls apart due to the absence of strong interactions between them (Fig. 4d, g). With the four corners fixed, the patch does not disassemble, although the crystals can still slip easily (Fig. 4e, h). The patch having all of its perimetric nodes fixed, on the other hand, shows excellent robustness and does not disintegrate (Fig. 4f, i).

To further quantify the stability of the woven crystalline patches, three-point bending tests were carried out on single crystals of **A** and **C** as well as on the crystalline patches **A2** and **C3** (Supplementary Movie 2). The stress was applied perpendicular to **A2** and along the weft, warp, and diagonal directions. The load borne by the patch after bending (Fig. 4j–l) was between 0.027 and 0.032 N (Fig. 4m). The load at the yield point of the weft was 0.036 N. The patch was then disassembled, and the fracture loads of the individual ten warp and weft crystals were found to be 0.0014–0.0041 N, with an average of 0.0023 N (Fig. 4n, o). These measurements show that the fracture strength of the crystalline patch is approximately 15.6 times that of the individual crystals. Similar to a conventional fabric, the crystals in the crystalline patch act cooperatively to dissipate the stress, thereby

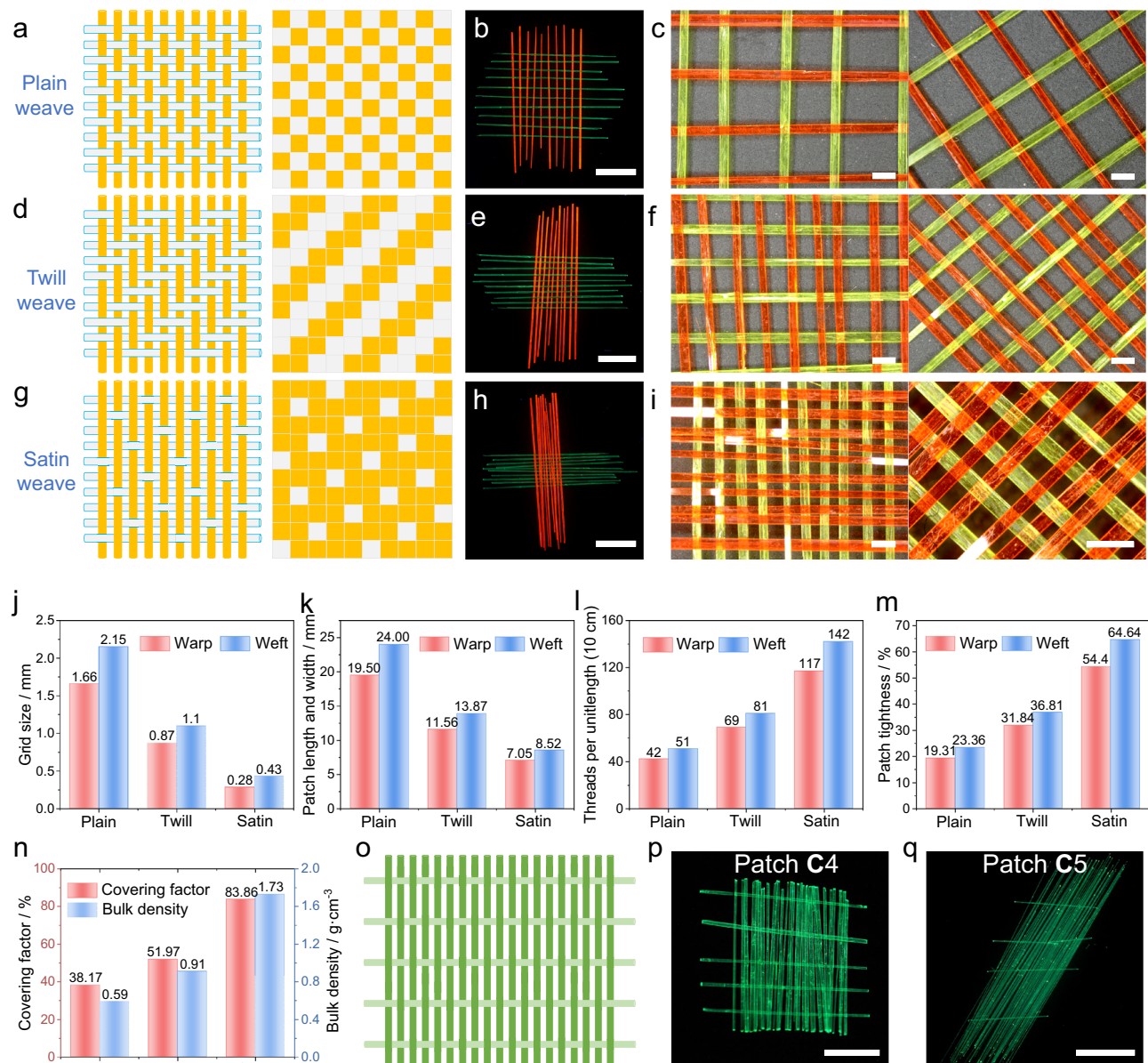

**Fig. 3 | A variety of crystalline weaving patterns. a, d, g** Schematic illustration of plain (**a**), twill (**d**), and satin (**g**) woven crystalline topologies. Yellow and gray (blue border) are used to show the difference between the topologies. Structural diagrams are shown on the left, organizational diagrams are on the right. **b, e, h** Fluorescence images of the corresponding crystalline patches (10 × 10), having a plain (**b**), twill (**e**), and stain (**h**) pattern. **c, f, i** Magnified photographs of the plain (**c**), twill (**f**), and stain (**i**) crystalline patches. **j–n** Comparison of different fabric parameters of plain, twill, and satin crystalline patches: average grid size (**j**), patch length and width (**k**), threads per unit length (**l**), patch tightness (**m**), coverage factor and bulk density (**n**). **o** Schematic of a crystalline patch having very disparate warp and weft densities. The green and light green lines represent the warp and weft crystals. **p, q** Fluorescence images of crystalline patches **C**4 and **C**5. The scale length in panels **p, e, h, p,** and **q** is 1 cm, and in panels **c, f** and **i** is 1 mm.

effectively enhancing the mechanical strength of the individual crystals. Considering that the outermost nodes of the fabric have been secured with glue, the interwoven network exhibits mechanical robustness that is borne by a cooperative stress-bearing mechanism. Certain crystals within the entangled network are expected to experience shear forces that typically surpass the magnitude of bending stresses, resulting in an increased bending rupture strength that is greater than the combined resistance of the individual, isolated crystals. The three-point bending tests were also performed on the crystalline patch **C**3 and the corresponding weft and warp crystals, and the mechanical properties follow a similar trend (Supplementary Figs. 14 and 15).

To test the effect of crystal thickness on the deformation ability of the woven crystalline patches, four crystalline patches of compound **C**

of different thicknesses and sizes were prepared. The average thicknesses of the crystals of **C** in the four patches were about 32, 64, 124, and 188 μm. Three-point bending tests were performed on the four crystalline patches (Supplementary Fig. 16a, b). It was concluded that with the increase in crystal thickness, the patch can sustain a larger load, while its ability to be deformed is decreased (Supplementary Fig. 16c). In addition, three-point bending tests were also performed on twisted crystals of **D** to study the contribution of these crystals to the fabric's overall plastic nature. As shown in Supplementary Fig. 17, compared to woven patches consisting of straight crystals of **D**, the patches made of twisted crystals of **D** are more susceptible to deformation. Twisted patches are more likely to develop shearing forces (stress on both flat faces of the crystal), and are more prone to fragmentation. Furthermore, to explore the effect of treatment with

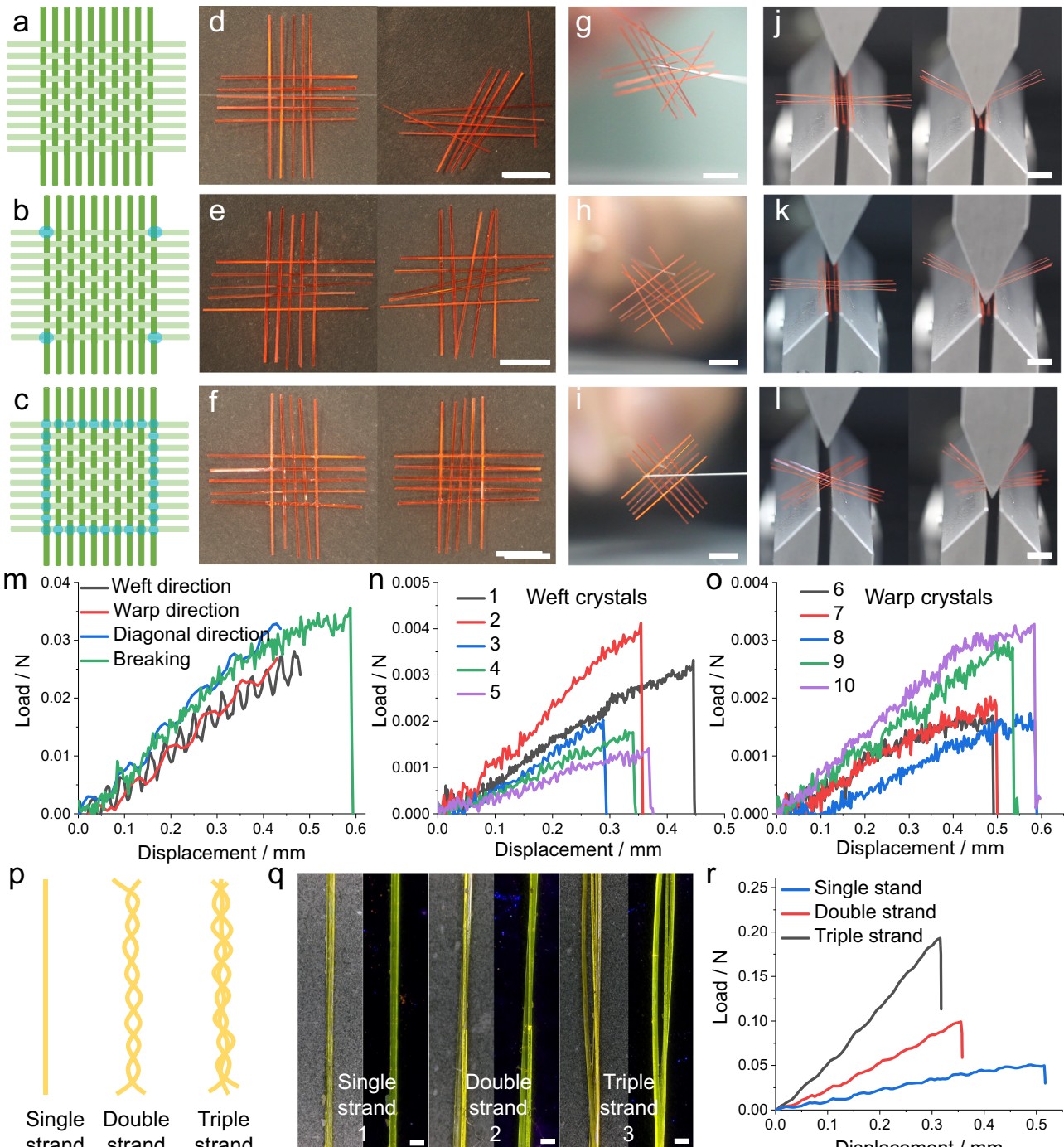

**Fig. 4 | Mechanical profile of the woven crystalline patches. a**–**c** Schematic of different forms of fixed crystal patch, including unfixed (**a**), fixed at four nodal corners (**b**), and fixed at multiple points along the perimeter (**c**). The green and light green lines represent the warp and weft crystals, and the blue dots represent the glue. **d**–**i** Photographs of different crystalline patches of **A** (5 × 5) before (left) and after (right) being swung violently (**d**–**f**), and then suspended in air (**g**–**i**). **j**–**l** Photographs of the three-point bending test of crystalline patch **A2**. Pressure is applied in the weft (**j**), warp (**k**), and diagonal directions (**l**). **m**–**o** Load-displacement curves obtained from the three-point bending test of crystalline patch **A2** (**m**) and the corresponding five weft crystals (**n**) and five warp crystals (**o**). **p** Schematic representation of single, double, and triple crystalline strands. **q** Photographs of crystals of **B** with single, double, and triple strands in daylight (left) and under UV light (right). **r** Load-displacement curves obtained from the three-point bending test of crystals of **B** having single, double, and triple strands. The scale length in panels **d**–**l** is 5 mm, and in the panel **q** is 100 μm.

boiling water on the mechanical properties of the crystals, we carried out three-point bending and tensile tests on crystals of **C** before and after soaking in boiling water (Supplementary Fig. 18). The bending moduli before and after the treatment were 4.56–4.81 GPa and 4.31–4.35 GPa, and the tensile moduli before and after treatment were 1.69–2.05 GPa and 1.57–1.88 GPa, respectively. The results indicate that

the crystals are thermally stable and the treatment does not have a significant effect on their mechanical properties.

The continuous fibrous material in a regular patch is generally a single yarn. If multiple strands of fiber are attached and twisted together, they form a multistrand yarn (Fig. 4p), with increased strength relative to the individual fibers. Crystals of **B** are long, thin, and soft,

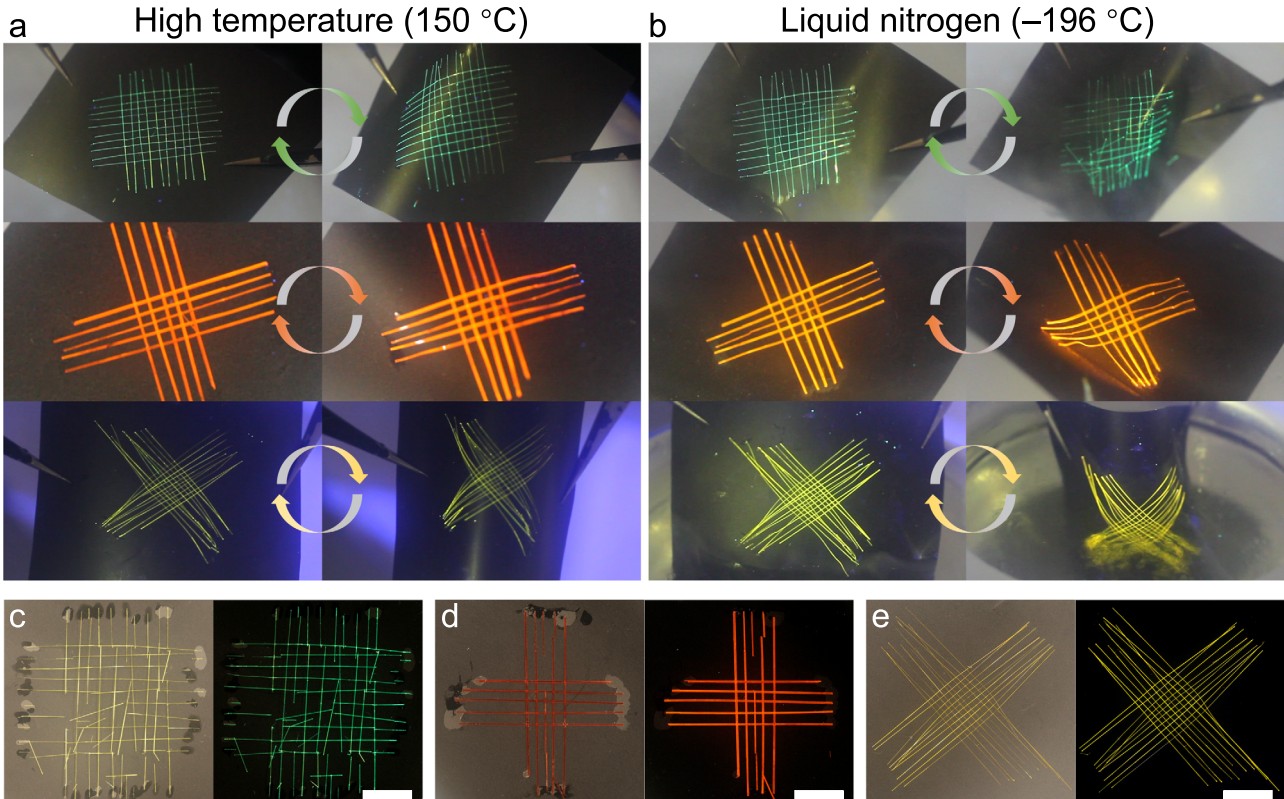

**Fig. 5 | Mechanical stability of the woven crystals under high and low temperatures. a, b** Bending-recovery process of three different patches at high temperature (**a**) and in liquid nitrogen (**b**). The woven crystalline patches were affixed to black paper, and then the paper was bent by applying pressure with tweezers. The arrows represent the straight and curved cycles. **c–e** Photographs of crystalline patches of **C** (**c**), **A** (**d**), and **B** (**e**) after alternate multiple bending at room temperature and low temperature recorded under daily light (left) and under UV light (right). Most of the crystals of patch **C** (10 × 10) were broken, patch **A** (5 × 5) was partially damaged, while patch **B** (10 × 10) remained intact. The scale length in panels **c–e** is 1 cm.

similar to fibers, and their Young's modulus was estimated to be 3.37 ± 0.20 GPa by three-point bending tests (Supplementary Fig. 19). Two or three elongated crystals of **B** were twisted together into double-stranded or triple-stranded crystal wires, respectively, and the two ends of the crystals were fixed (Fig. 4q; Supplementary Fig. 20). The linear correlation between the load and displacement demonstrated that the bending process is elastic for single-, double- and triple-stranded crystals of **B** (Fig. 4r). Moreover, as the number of crystal strands increases, the load increases for the same displacement during three-point bending, revealing that multistrand crystals have enhanced mechanical strength.

The mechanical robustness and stability of materials and devices are central during operations in extreme or harsh environments, such as very high or low temperatures[49], and this prompted us to investigate the thermal stability of the woven crystalline patches. Three different crystals were employed to prepare woven patches of **C** (10 × 10), **A** (5 × 5), and **B** (10 × 10). To explore the flexibility of crystalline patches at high and low temperatures, they were fixed on black paper and placed in an oven at 150 °C for 6 h. The patches were taken out and bent several times, while using tweezers to hold the black paper, without damage (Fig. 5a). The thermal stability of the crystals was also determined (Supplementary Fig. 21). The woven structures remain flexible at high temperatures below the melting points of **A**, **B**, and **C** (205, 197, and 227 °C, respectively), and we suggest that this is a more general principle (unless the compounds undergo phase transitions). They were then bent several times over liquid nitrogen, immersed in the liquid nitrogen, and bent several times again (Fig. 5b; Supplementary Movie 3). As shown in Fig. 5c–e, after bending in liquid nitrogen, while most of the crystals in patch **C**

have broken up and one crystal in patch **A** was also fractured, the patch made of crystals of **B** remained undamaged. This behavior indicates that woven structures with different crystal compositions and mechanical properties of the component crystals have different levels of flexibility at low temperatures (Supplementary Figs. 22–24). Mechanically flexible and robust crystalline patches can be prepared by selecting crystals that remain flexible at both high and low temperatures.

Periodic micro- and nanoarrays of transmissive elements have been used for microelectronic circuits, imaging devices, and sensors. However, such applications typically require materials that are compatible with the traditional microfabrication techniques, which has to some extent limited the interest in thin-film-based arrays[50]. Organic crystals have been suggested as optical waveguiding media recently[22,27,32,51], and thus we embarked on exploring the performance of the woven crystalline patches as optical waveguiding arrays. The optical waveguiding capability of single crystals of **A**, **B**, and **C** were characterized for crystals of lengths of 5 mm and 1.5–3.0 cm, and the corresponding optical loss coefficients (OLCs) were found to be 0.41, 0.20, and 0.29 dB mm⁻¹ for the 5 mm length crystal (Supplementary Fig. 25). When the crystal length was of centimeter-scale, a small increase in the optical loss coefficient was observed (Supplementary Fig. 26), indicating that the long crystals used for weaving retain commendable optical transmission properties. Furthermore, to evaluate the effect of weaving on the optical transmission performance of crystals, the same tests were performed on six crystals (the warp and weft crystals **A**, **B**, and **C**) of the patch **A**#**B**#**C**. The OLC values of the woven crystals are close to the aforementioned values for crystals **A**, **B**, and **C** (Supplementary Fig. 27). As shown in Fig. 6a, the warp and weft

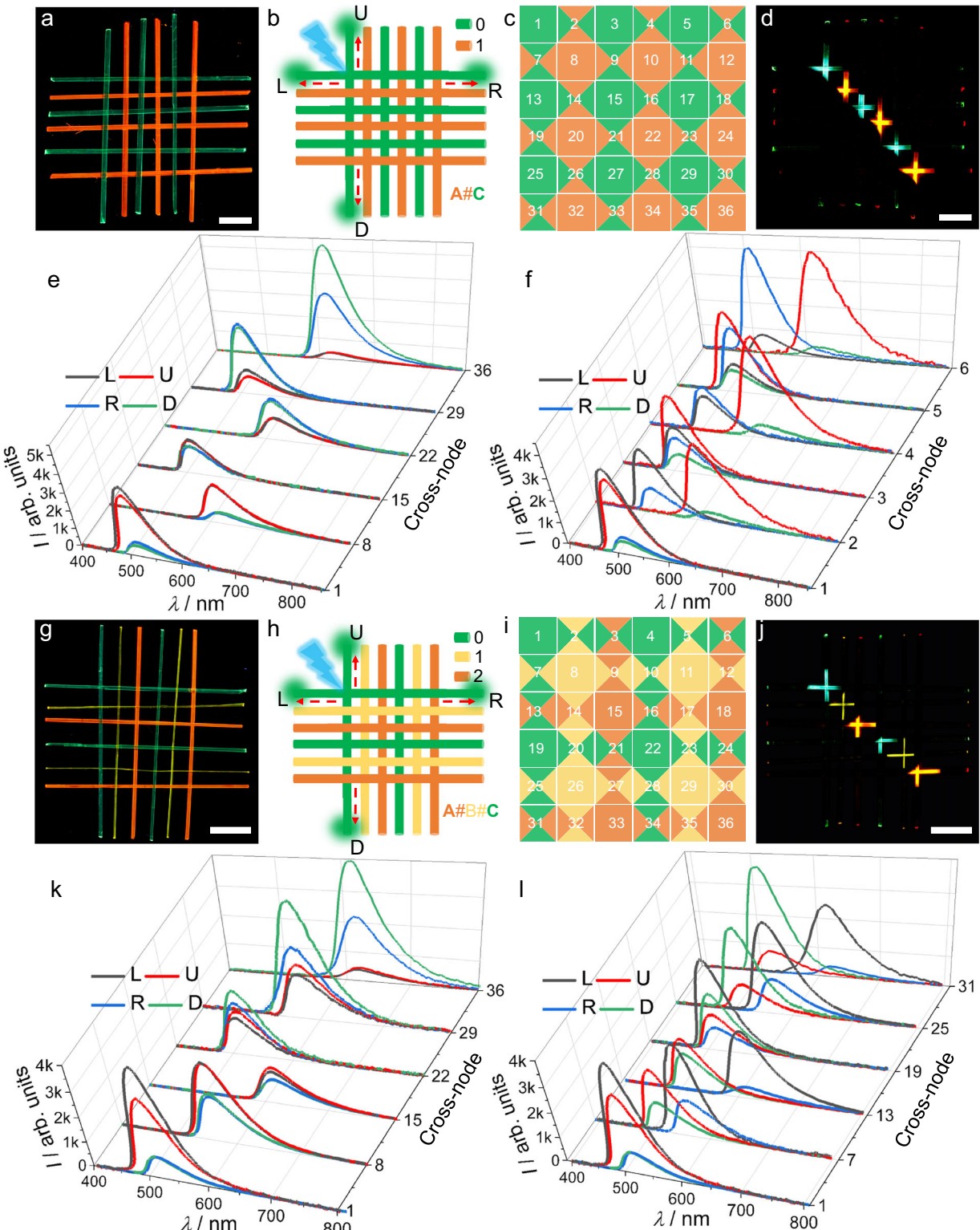

**Fig. 6 | Optical transmission through heterogenous woven crystalline patches.**
**a**, **g** Fluorescence images of crystalline patches **A#C** (**a**) and **A#B#C** (**g**).
**b**, **h** Schematic of the optical waveguide arrays based on **A#C** (**b**) and **A#B#C** (**h**).
The orange-red, yellow and green lines represent crystals **A**, **B**, and **C**, respectively.
The green highlight indicates the optical waveguide signal output. The blue light
represents excitation by a 355 nm laser, and the red dashed arrow indicates the
direction of light transmission. **c**, **i** Schematic of the output signals of optical
waveguide arrays based on **A#C** (**c**) and **A#B#C** (**i**). The four-part division of
the square represents the output signal in the four directions (L, U, R, and D).

The different colors represent the fluorescence emission of the output signal. The
patches' cross-nodes are designated by numbers 1–36. **d**, **j** Crystalline patches
**A#C** (**d**) and **A#B#C** (**j**) were excited with a 355 nm laser focused at different posi-
tions from left to right along the diagonal direction. The scale length in panels
**a**, **d**, **g**, and **j** is 2 mm. **e**, **f** Emission spectra collected at the four ends of the patch
**A#C** at various excitation positions (along the diagonal direction (**e**) and from left to
right (**f**)). **k**, **l** Emission spectra collected at the four ends of **A#B#C** and at various
excitation positions (along the diagonal direction (**k**) and from top to bottom (**l**)).

crystals of the patch **A**#**C** (6 × 6) are composed of alternating crystals of **A** and **C**. To avoid the excitation of adjacent crystals when woven organic crystal wires are stimulated point-to-point, a black paper mask with a pinhole of the size of the laser beam was applied to cover the crystals (Supplementary Figs. 28 and 29).

When the first interlacing point (node) of **A**#**C** was excited at 355 nm, green emission with different intensities was transmitted along the four directions (referred to as left (L), up (U), right (R), and down (D); Fig. 6b). If we define the green light signal as 0 and the orange light signal as 1, then the combination of signal outputs in the four directions (L U R D) from this first cross-node is (0 0 0 0). If the 36 cross-nodes in **A**#**C** are numbered successively 1–36 (Fig. 6c), then a regular arrangement of signal outputs (0 0 0 0), (0 1 0 1), (1 0 1 0), and (1 1 1 1) can be obtained (Supplementary Fig. 30). For instance, when five cross-nodes (1, 8, 15, 22, 29, and 36) were orderly excited at 355 nm along the diagonal direction, the fluorescence in all four directions remained consistent, and the output signal alternated between (0 0 0 0) and (1 1 1 1) (Fig. 6d). The intensity across these nodes gradually increased in the L and D directions and diminished in the U and R directions, which is consistent with the variation in distance between the excitation (interlacing) and output points (Fig. 6e). The output signals from the four directions of the 36 cross-nodes were collected (Supplementary Figs. 31 and 32), and the distances between the nodes and the four outputs were also measured (Supplementary Table 10). When nodes 1–6 were excited sequentially from left to right, the signal fluorescence of the outputs L and R remained unchanged as green emission, while the intensity decayed/enhanced exponentially. On the other hand, the signal intensity of the outputs U and D did not change significantly, while the fluorescence emission alternated between green and orange (Fig. 6f).

We also wove three different types of crystals to obtain a 6 × 6 crystalline patch, **A**#**B**#**C** (Fig. 6g). As shown in Fig. 6h, i, compared with the patch composed of two crystals, **A**#**B**#**C** displays higher complexity in the optical waveguide array (Supplementary Fig. 33). The corresponding spectral and distance data at the 36 nodes were also collected (Supplementary Figs. 34 and 35; Supplementary Table 11). When the nodes 1, 8, 15, 22, 29, and 36 were excited along the diagonal direction, the fluorescent colors of the signals at the four outputs of **A**#**B**#**C** remained consistent as in the case of **A**#**C**, but the color of the output light alternated between green, yellow, and orange (Fig. 6j, k). When nodes 1, 7, 13, 19, 25, and 31 were excited in turn from the top to the bottom, the fluorescence emission of the outputs L and R alternated between green, yellow, and orange light (Fig. 6l). The signal fluorescence of the outputs U and D remains unchanged as green emission, while the intensity decayed or enhanced exponentially. To our knowledge, these interlaced, ordered arrays of optically transmissive crystals are the first of their kind; the output signal depends on the excited node, and we envisage applications in optoelectronics for information integration and encryption, among other potential applications.

With the possibility of having both active and passive waveguiding, we designed multi-mode optically transmissive networks based on the woven crystals and radiation wavelengths. Red, yellow, and green fluorescence signals were denoted 0, 2, and 1, respectively. Upon excitation at 355 nm, 654 nm, or with a combination of two wavelengths (355 and 654 nm), a single crystal of **A** always emitted red light (0) (Fig. 7a–c). On the contrary, with the same excitations, a single crystal of **C** emitted green (1), red (0), and yellow (2) light (Fig. 7d–f). As shown in Fig. 7g, i, and k, three different optically transmissive networks (denoted mode-1–3) were constructed based on woven patches of **A** and **C**. The nodes of these nets were excited by using lasers of different wavelengths and the resulting emission was observed (Supplementary Figs. 36–38). In the case of mode-1 which comprises only crystals of **C** (Fig. 7g), regardless of the excited node, the output signals are identical in all directions and change with the excitation

wavelength in the same way as those of a single crystal of **C** (Fig. 7h). In the optical network mode-2 all crystals of the warp are **A**, while all crystals in the weft are **C** (Fig. 7i). Because at each node two different crystals are interlaced, the output signals are independent of the node of excitation, although they vary across the output directions (Fig. 7j). In the up (U) and down (D) directions the output is always 0, while in the left (L) and right (R) directions the output varies with the excitation wavelength. In the case of mode-3, both the weft and the warp are composed of alternating crystals of **A** and **C** (Fig. 7k). In this particular case, when the patch is excited at 355 nm or at both 355 nm and 654 nm (Figs. 7l, m), the output pattern varies across the nodes. For example, the signal outputs of nodes 1, 3, 7, and 9 are consistent with mode-2, while the outputs of nodes 2 and 8 are consistent with mode-1. In the case of nodes 4 and 6, the output does not depend on the excitation and is always 0. We envision that this combination of the optical waveguiding capability of organic crystals and the deterministic dependence of the optical output on the topology of the network could be applied in the future to spatially resolved optical sensing, among other prospective applications.

In summary, we have developed a weaving method for the preparation of patches of flexible organic single crystals that resemble common textiles. The method is not limited to the type of crystals, and we succeeded in preparing a wide variety of crystalline patches, both from the same and from different flexible organic crystals. The crystals, both plain and twisted, were woven in plain, twill, and satin topologies. The patches are about 15 times more resilient to failure compared to individual crystals, which reflects their enhanced collective firmness. The thermal stability of this crystalline fabric depends on the crystals' resilience to high or low temperatures. By selecting the crystals, the patches remain flexible between 77 K and 423 K. Moreover, we present an optical array model based on the woven crystal waveguides by exploiting the regular arrangement of the crystals in a hybrid patch. Within a more general context, the crystalline patches provide a way to extend 1D crystals into flexible, integrated 2D planar structures, with potential future applications in flexible electronics ranging from sensing devices to optical arrays.

## Methods
### Materials and synthesis
The solvents (purity: ≥ 99.5 %) and all starting materials (purity: ≥ 97 %) for the organic syntheses were obtained from Shanghai Titan Scientific and Energy Chemical, respectively, and were used as received. Compounds **C** (purity: ≥ 97.5 %) and **D** (purity: ≥ 97.0 %) were purchased from Energy Chemical and used without further purification. Compounds **A** and **B** were synthesized according to procedures described previously[26,46]. Compound **A**: $^1$H NMR (400 MHz, DMSO-$d_6$) $\delta$ = 12.64 (s, 1H), 8.95 (s, 1H), 8.04 (s, 1H), 7.94 (s, 1H), 7.81 (d, $J$ = 8.1, 2H), 7.56–7.45 (m, 3H), 7.25 (d,$J$ = 8.5, 1H), 7.17 (d,$J$ = 3.4, 1H), 6.89 (d,$J$ = 8.4, 1H), 6.82–6.75 (m, 1H), 2.28 (s, 3H). $^{13}$C NMR (126 MHz, DMSO-$d_6$) $\delta$ = 164.05, 158.63, 150.20, 149.10, 146.84, 134.80, 132.73, 132.15, 128.51, 128.26, 126.97 (2 C), 122.68 (2 C), 119.49, 118.01, 117.61, 116.97, 113.70, 105.56, 20.40. Compound **B**: $^1$H NMR (400 MHz, DMSO-$d_6$) $\delta$ = 8.27–8.20 (m, 2H), 8.14 (d, $J$ = 2.1, 1H), 8.11–7.99 (m, 3H), 7.61–7.43 (m, 3H), 6.96 (dd, $J$ = 8.9, 2.4, 1H), 3.86 (dd, $J$ = 15.9, 2.1, 6H). $^{13}$C NMR (101 MHz, DMSO-$d_6$) $\delta$ = 188.19, 155.91, 142.62, 140.25, 137.53, 133.53, 133.20 (2C), 129.19 (2 C), 127.04, 118.92, 114.82, 114.70, 112.67, 112.26, 112.06, 103.41, 56.14, 33.83.

### Crystal growth
The saturated solution in dichloromethane (DCM) of compound **A** (~18.5 mmol L$^{-1}$) was placed in a round-bottom flask, and a layer with approximately twice the volume of ethanol was carefully added along the wall while taking care not to disturb the first layer. After allowing for a slow diffusion to occur by keeping the crystallization mixture at room temperature, needle-shaped crystals of **A** were obtained in two

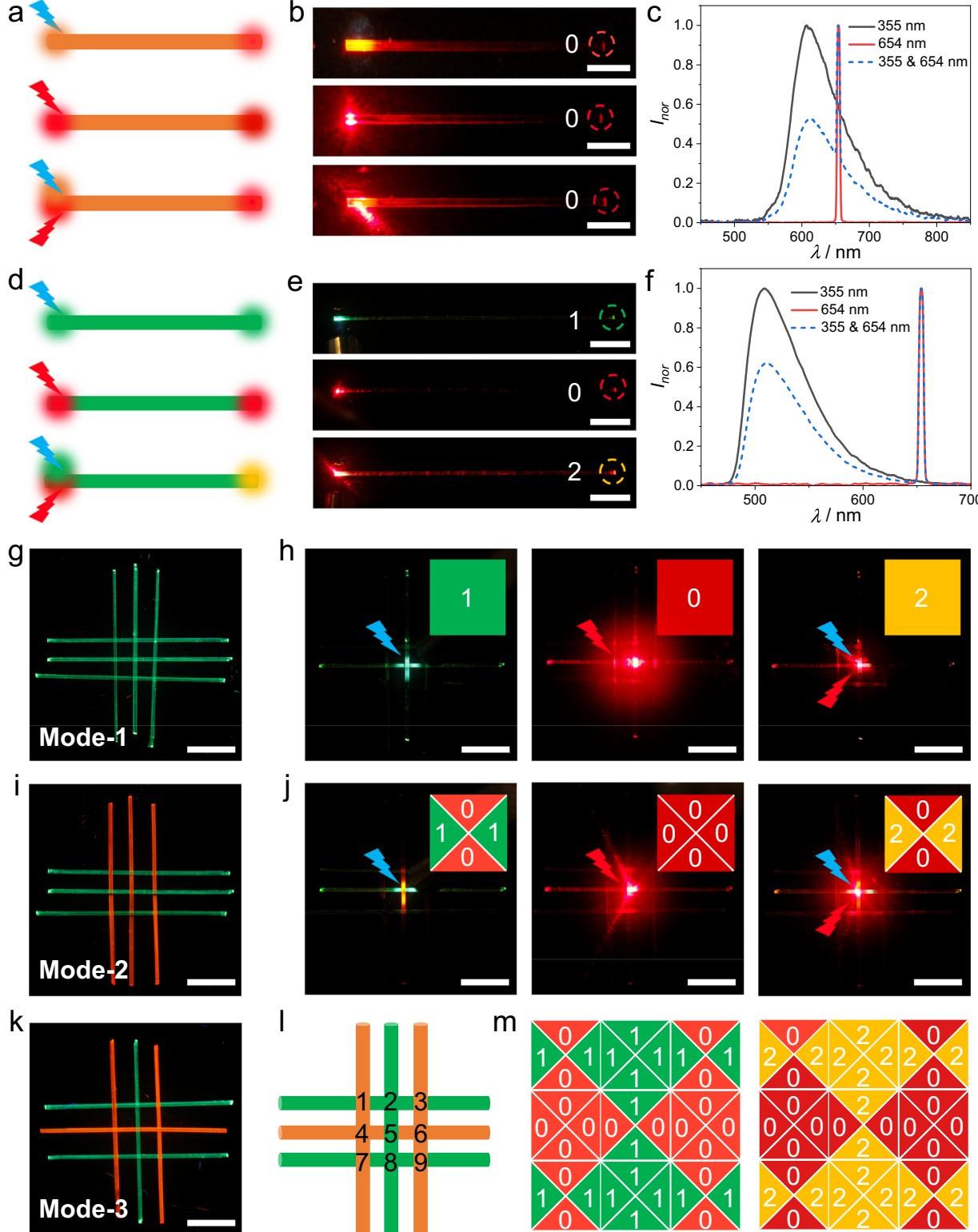

**Fig. 7 | Optically transmissive networks based on woven emissive crystalline patches. a, d** Illustration of the dependence of the emissive output from crystals **A** (**a**) and **C** (**d**) excited at 355 nm (blue arrow), 654 nm (red arrow), as well as at 355 and 654 nm. Different color highlights indicate signals in active or passive optical waveguides of different crystals. **b, e** Photographs of actual crystals of **A** (**b**) and **C** (**e**) excited with lasers of different wavelengths. The images correspond to the schemes in panels **a** and **d**. **c, f** Spectra collected at the tip or middle of crystals **A** (**c**) and **C** (**f**) excited with lasers of different wavelengths. **g, i, k** Fluorescence images of crystalline patches used as "optically transmissive networks" composed of crystals of **C** (mode-1, **g**), **A** and **C** with all warp crystals of **A** and all weft crystals of **C** (mode-2) (**i**), and

**A** and **C** with alternating crystals of each kind (mode-3) (**k**). **h, j** Patches of mode-1 (**h**) and mode-2 (**j**) excited with lasers of different wavelengths (from left to right, the excitation is 355 nm, 654 nm, and 355/654 nm). The insets show a color-coded representation of the output signal in four directions ('1' for green, '0' for red, and '2' for yellow light). **l** Schematic of mode-3 based on the patch shown in panel **k**. The orange red and green lines represent crystals **A** and **B**, and the numbers are ordinal numbers of the nine nodes. **m** A schematic of logical operation based on mode-3. The triangles represent the output signal in the four directions, left (L), up (U), right (R), and down (D). The different colors and numbers represent the type of output signals. The scale length in panel **b** is 2 mm, and in panels **g–k** it is 4 mm.

days. Crystals of **B**–**D** were obtained in a similar way, by adding ethanol on top of dilute solutions of **B**–**D** dissolved in DCM (concentrations: ~6.3 mmol L$^{-1}$, ~6.0 mmol L$^{-1}$, and ~8.9 mmol L$^{-1}$ for **B, C,** and **D**, respectively).

## X-ray crystallographic analysis

Diffraction data of crystals **C** before and after immersion in boiling water were collected on a Bruker D8 Venture diffractometer. The data collection, integration, scaling, and absorption corrections were performed by using the Bruker Apex 3 software[52]. The structures were solved by using direct methods using Olex2[53], and refined by using the full-matrix least-squares method on $F^2$. The non-hydrogen atoms were refined anisotropically, while the positions of the hydrogen atoms were calculated and refined isotropically. The crystallographic details are available as Supplementary Table 2. The crystallographic information has been deposited at the Cambridge Crystallographic Data Centre (CCDC) with the ID 2289561 for both the original crystal of **C**, and for the crystal of **C** after immersion in boiling water (only one CCDC number is provided for essentially identical crystal structures).

## Optical waveguiding tests

The sample was irradiated with the output of the third harmonic (355 nm) of a Nd:YAG (yttrium aluminum garnet) laser, with a pulse duration of about 5 ns. The energy of the laser was adjusted by using calibrated neutral density filters. The beam was focused onto a strip whose shape was adjusted to 0.5 × 0.5 mm by using a plano-convex lens and a slit. The crystal was placed on a silicon wafer, and one tip of the crystal extended out of the edge of the wafer to align with the probe of the spectrometer. While changing the location of excitation, spectral data was collected for each irradiated location at the excitation site and at the tip of the crystal. All emission spectra were recorded on a Maya2000 Pro CCD spectrometer. For the array waveguiding tests of the crystalline patches, as shown in Supplementary Figs. 28 and 29, the patch was placed on a silicon wafer and marked to ensure that the distance between the output end and the optical fiber remained constant when the patch was moved. By moving the patch, different nodes were excited by the laser in turn, and the corresponding output emission spectra of the fabric were collected in one direction (L). Then the patch was rotated to align the other directions (U, R, and D) with the optical fiber, and the output signals from other nodes in each direction were collected.

## Data availability

The X-ray crystallographic coordinates for structures reported in this study have been deposited at the Cambridge Crystallographic Data Centre (CCDC), under deposition number 2289561. These data can be obtained free of charge from The Cambridge Crystallographic Data Centre via www.ccdc.cam.ac.uk/data_request/cif. All relevant data generated in this study have been deposited in the Figshare database under the accession code https://doi.org/10.6084/m9.figshare.24407752.v2. Source data are provided in this paper.

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

## Acknowledgements

This work was supported by the National Natural Science Foundation of China (52173164, H.Z.) and a fund from New York University Abu Dhabi (P.N.). This material is based upon works supported by Tamkeen under NYUAD RRC Grant No. CG011 (P.N.).

## Author contributions

L. Lan, J.Q., X.P., and Q.D. performed the experiments. L. Li, P.N., and H.Z. supervised the experiments. H.Z. and P.N. conceived the project.

## Competing interests

The authors declare no competing interests.
