## [Peer Review File · Nature Communications]

REVIEWER COMMENTS

Reviewer #1 (Remarks to the Author):

The work in this manuscript presents an interesting application of mechanically flexible emissive organic single crystals. Authors have utilized the exceptional mechanical flexibility of organic crystals to make woven 2D networks of different patterns using multiple types of organic materials. The results are extremely interesting and can be of great benefit to the future generation optical waveguides, electronics, etc. The work is done thoroughly explaining all the required details. The manuscript can be accepted after a minor revision, after addressing the below comments.

1. It is interesting to note that the strength of the woven network is greater than the sum of the individual crystals. Some details on how this works will be helpful.

2. What could be the other applications of the woven networks? Will they be useful for flexible electronics. Recently some examples of elastic crystals for electronic applications by the groups of Gong, Reddy, etc, (not cited) have been reported. Do you see similar type of applications with these networks?

3. The details of the twisted crystals are not discussed in the article (while some images in the SI are presented). Some comments in the main draft would be useful.

Otherwise, I am happy with the manuscript.

Reviewer #2 (Remarks to the Author):

The authors used hand weaving techniques to weave elastic organic crystals into simple logic devices. Although this method is somewhat innovative, such as weaving organic crystals, its universality and applicability are insufficient. Hence, I would not recommend it to be published in Nature Communications. Specific comments are as follows, hope they are useful for the authors to further improve the quality of the work.

1. In terms of material selection, organic crystals with plasticity and toughness are required, and they must be prepared into cm-sized crystal wires. In this process, precise control of the length, thickness, and width of the crystal wires is required. The authors did not elaborate the solution to abovementioned issues, so in my opinion, this method is not universal.

2. Even if the technology mastered by the authors can achieve the controllable growth of organic crystal wires, long crystal wires will inevitably encounter problems with numerous crystal defects, as shown in Supplementary Figure 10. Defects will cause huge optical losses during light transmission, which is not conducive to related optoelectronic applications. To this end, the authors should clearly explain the solution to related technical problems in detail and provide relevant scanning electron microscope or fluorescence microscope images of their crystal wires.

3. The authors did not study the luminescence intensity and luminescence efficiency of organic crystal wires. Similarly, if there are many defects in organic crystal wires, it is difficult to guarantee good light transmission effects, and the luminescence efficiency will decrease as the length increases.

4. The authors made the woven organic crystals into optically transmissive webs. Its luminescence mechanism is photoluminescence. When tightly woven organic crystal wires are stimulated point-to-point, adjacent parts of organic crystal wires are also stimulated to emit light, as shown in Figure 7h and 7i. How to avoid interference from other signals?

5. Hand weaving also has limitations. Crystal wires that are too short are not conducive to weaving, and long crystal wires may encounter challenges in luminescence efficiency and light transmission effects due to numerous defects during crystal growth. The authors should consider how to resolve this dilemma.

Reviewer #3 (Remarks to the Author):

I have carefully reviewed the manuscript "Woven organic crystals" and find the work to be a promising step toward harnessing the benefits of flexibility in molecular crystals for advanced applications. The manuscript presents a well-written study with a proposed strategy to address the identified challenges. However, to further enhance the acceptability of your study for practical applications, I recommend considering the following points:

1. As this work serves as a prototype, it is crucial to outline your intentions for future advancements. Clearly define how you plan to refine and expand your study to make it more applicable to real-world scenarios and industrial applications.
2. In most assemblies, the porosity and thickness of crystals tend to be arbitrary, unlike the well-controlled molecular weaving observed in crystals. Investigate how these random distributions affect both the mechanical and luminescence properties of the crystals. Specifically, explore how increasing crystal thickness influences the tendency for elastic crystals to undergo plastic deformation under higher stress.
3. On page 5, line 121, you mention that "the crystal density is inversely correlated with the crystal thickness." Please provide a clear explanation and supporting evidence for this statement to validate its accuracy.
4. The treatment of the crystalline fabric with boiling water to remove the glue from the weft of the porous structures is an important step. However, assess the potential detrimental effects of this treatment on crystallinity. High temperatures during this process can lead to thermal agitation of molecules, potentially causing disorder in the crystals.
5. Given that the assemblies consist of different crystals with varying degrees of elastic flexibility and distinct melting temperatures, it becomes crucial to establish correlations between their mechanical properties after the treatment with boiling water. Provide the homologous temperature of the crystal system to better illustrate the possible thermal effect of this treatment.
6. Apart from plain weaving, incorporating twisted crystals into the fabric introduces two-dimensional plasticity in addition to elastic properties. Investigate how such crystals contribute to the fabric's overall plastic nature, particularly under higher stress conditions.

In conclusion, the manuscript shows significant promise in exploring the potential of flexible woven organic crystals for various applications. Addressing the above-mentioned comments will enhance the completeness and applicability of the study. I recommend revising the manuscript in light of these suggestions, and I look forward to seeing your work published after these improvements.

Response to the comments from Reviewer #1:

Overall Comment: *The work in this manuscript presents an interesting application of mechanically flexible emissive organic single crystals. Authors have utilized the exceptional mechanical flexibility of organic crystals to make woven 2D networks of different patterns using multiple types of organic materials. The results are extremely interesting and can be of great benefit to the future generation optical waveguides, electronics, etc. The work is done thoroughly explaining all the required details. The manuscript can be accepted after a minor revision, after addressing the below comments.*

Comment: 1. *It is interesting to note that the strength of the woven network is greater than the sum of the individual crystals. Some details on how this works will be helpful.*

Response: We thank the Reviewer for the careful reading the manuscript and their positive comments. In the revised version, we addressed all of their comments. In response to this comment, a brief discussion explaining the reasons for the higher mechanical strength of the woven network has been added in the introduction (Page 8, Line 208).

Added text: “Considering that the outermost nodes of the fabric have been secured with glue, the interwoven network exhibits mechanical robustness born by a cooperative stress-bearing mechanism. Certain crystals within the entangled network experience shear forces that typically surpass the magnitude of bending stresses, resulting in an increased bending rupture strength that surpasses the combined resistance of the individual crystals.”

Comment: 2. *What could be the other applications of the woven networks? Will they be useful for flexible electronics. Recently some examples of elastic crystals for electronic applications by the groups of Gong, Reddy, etc, (not cited) have been reported. Do you see similar type of applications with these networks?*

Response: We thank the Reviewer for bringing these works to our attention. We have added the relevant articles in the revised manuscript. In 2021, Gong and collaborators provided the first example of deformation-induced function enhancement in crystal materials (*Angew. Chem., Int. Ed.* 2021, 60, 22424–22431). The bending crystal devices show a significant increase beyond seven orders of magnitude in conductivity compared to the straight ones. In 2023, Reddy and the collaborators demonstrate a fully flexible field effect transistor (FET) with methylated diketopyrrolopyrrole (DPP-diMe) crystals (*Chem. Sci.* 2023, 14, 1363–1371). The flexible FET retained field effect mobility for up to 40 bending cycles without any decrease of mobility. Woven optoelectronic textiles have already proven their potential across a spectrum of applications: wearable supercapacitors, sensors, storage devices, light-emitting diodes, and transistors (Page 1, Line 33) have all been prepared. In our research, we aimed at exploring a variety of properties of the crystalline frameworks, possibly by combining them with other materials. This innovative approach unlocks the potential for dual-mode information transmission through the photoelectric network, while it also paves the way for development of logic circuits based on organic crystals. We hope to explore that aspect in the near future.

Accordingly, the following text has been added (Page 2, Line 57) in the revised manuscript:

Added text: “Inspired by some other examples of elastic crystals that have been recently utilized for electronic applications,^[44,45] our future research endeavors will include the fabrication of composite woven crystal networks with conductive capabilities which are

expected and designed to function as logic circuits.”

Comment: 3. *The details of the twisted crystals are not discussed in the article (while some images in the SI are presented). Some comments in the main draft would be useful.*

Response: We agree with the Reviewer. To respond to this comment, the following text has been added (Page 5, Line 129) in the revised manuscript:

Added text: “The crystals in the patches of **D** were twisted along their longest crystal axis. The length of the helical pitch of all crystals of **D** ranged from 1.06 to 1.51 mm.”

Response to the comments from Reviewer #2:

Overall Comment: *The authors used hand weaving techniques to weave elastic organic crystals into simple logic devices. Although this method is somewhat innovative, such as weaving organic crystals, its universality and applicability are insufficient. Hence, I would not recommend it to be published in Nature Communications. Specific comments are as follows, hope they are useful for the authors to further improve the quality of the work.*

Comment: 1. *In terms of material selection, organic crystals with plasticity and toughness are required, and they must be prepared into cm-sized crystal wires. In this process, precise control of the length, thickness, and width of the crystal wires is required. The authors did not elaborate the solution to abovementioned issues, so in my opinion, this method is not universal.*

Response: We thank the Reviewer for the careful reading and all the comments, which we tried to address to the best of our ability. In this manuscript, we present a highly innovative exploratory approach to the preparation of two-dimensional materials. The crystals that are required for weaving, indeed, need to be elongated, as is the case with the yarns used to make textile. The resulting patches of conventional fabrics do require centimeter-sized and needle-like crystals with certain elasticity and toughness. However, while the process of preparation of the crystals is out of the scope of our proposed weaving method, we note that there is an abundance of crystals in the literature that have been described to naturally grow in such habit. As shown in Figure 2, woven patches can be fabricated from crystals with slightly different thickness and width, and therefore, a precise control of the crystal's dimensions is not required. Many long and thin fluorescent organic crystals have been reported (for some examples, see: *Angew. Chem. Int. Ed.* 2018, 57, 8448–8452; *Angew. Chem. Int. Ed.* 2018, 57, 17002–17008; *Adv. Opt. Mater.* 2021, 9, 2002264; *Adv. Opt. Mater.* 2020, 8, 2000959; etc.), and by definition, for the crystals to be tested as elastic or olastic, they need to be elongated; blocky crystals cannot be bent. So the ‘universality’ of the approach that we describe here naturally does not refer to all crystals, but to organic crystals which are available as long habits.

Accordingly, the following text has been added (Page 4, Line 118) in the revised manuscript:

Original text: “The weaving is not restricted by the chemical composition of the crystals as long as they are slender and elastic.”

Revised text: “The availability of crystals with elongated habit via the established crystal growth techniques poses a notable challenge, where the experimenter has a limited control.

However, the proposed weaving method is not restricted by the chemical composition, width, and thickness (generally within a few hundred microns) of the crystals, as long as they are slender (centimeter size) and reasonably elastic.”

Comment: 2. *Even if the technology mastered by the authors can achieve the controllable growth of organic crystals wires, long crystal wires will inevitably encounter problems with numerous crystal defects, as shown in Supplementary Figure 10. Defects will cause huge optical losses during light transmission, which is not conducive to related optoelectronic applications. To this end, the authors should clearly explain the solution to related technical problems in detail and provide relevant scanning electron microscope or fluorescence microscope images of their crystal wires.*

Response: In the course of this work, thousands of long needle-like crystals were obtained by using controllable growth methods, which were then screened by morphology and optical transmission properties (using a microscope and optical waveguide devices). Eventually, a certain number of crystals with fewer defects that can be used for optoelectronic applications were obtained (see new Supplementary Figure 2). Crystals **B** shown in Supplementary Figure 10 have a fewer defects, and are attached and twisted together to form multi-strand yarns. Following the suggestion by the Reviewer, in the revised manuscript the related scanning electron microscopy and fluorescence microscopy images of four crystals have been provided (new Supplementary Figures 2 and 3).

The following text and new Supplementary Figures 2 and 3 have been added (Page 3, Line 74) in the revised manuscript and the Supporting Information:

Added text: “In the course of this work, thousands of long needle-like crystals were obtained by using controllable crystal growth methods, which were then screened for their morphology and optical transmission properties. From these, a certain number of crystals that can be used as optical waveguides have been selected (Supplementary Figure 2). These crystals are relatively uniform in size and have a smooth surface, which suggests high quality with only a small number of defects (Supplementary Figures 2 and 3). We believe that in a future application, the process of selection of crystals of sufficient quality can be automated and therefore more efficient.”

New Supplementary Figure 2. (a, b) Photographs of the as-grown (top) and selected (bottom) crystals **A** and **C** in daylight (top) and under UV light (bottom). (c) Photomicrographs of crystals **A–D** in daylight (left) and under UV light (right). The scale bar in panels a and b is 2 cm, and in panel c is 500 μm .

New Supplementary Figure 3. Scanning electron micrographs showing surfaces of crystals **A–D**. The scale bar in all panels is 100 μm .

Comment: 3. *The authors did not study the luminescence intensity and luminescence efficiency of organic crystal wires. Similarly, if there are many defects in organic crystal wires, it is difficult to guarantee good light transmission effects, and the luminescence efficiency will decrease as the length increases.*

Response: We thank the Reviewer for this comment. Indeed, it would be important to study the luminescence efficiency of organic crystal wires. Accordingly, the fluorescence quantum yields were measured for crystals **A–D** and the results were added as new Supplementary Table 1 in the revised version. In relation to the effect of crystal length on the luminescence efficiency, we investigated the fluorescence quantum yields of short crystals (shorter than 1 mm) and long crystals (~1 cm). The results showed that the quantum yields of long crystals was reduced slightly compared to those of small crystals.

The defects of the crystal are expected to have a certain effect on its optical waveguide ability, resulting in increased optical loss. Therefore, we also performed optical waveguiding tests on crystals of **A–C** with a length of 1.5 – 3.0 cm (new Supplementary Figure 21). Compared with optical loss coefficients (OLCs) of crystals **A–C** at length of 5 mm, the OLCs of long crystals **A–C** were only slightly higher. This shows that the long crystals used for weaving still have acceptable optical waveguide properties.

To clarify this point, the following text (Page 3, Line 69) and new Supplementary Table 1 have been added in the revised version of the manuscript and the Supporting Information:

Added text: “The fluorescence quantum yields of the four types of crystals were measured for both short (less than 1 mm) and long (about 1 cm) crystals (Supplementary Table 1). The results showed that the quantum yields of the longer crystals were slightly reduced compared to those of the short crystals, an observation that underscores the minor impact of defects present in the longer crystals on their overall luminescent properties.”

New Supplementary Table 1. Emission peak maxima (λ_{em}) and quantum yields (Φ) of crystals of **A–D** of various sizes

Crystal	Length	λ_{em} (nm)	Φ (%)
A	< 1 mm	595	17.33
	~ 1 cm		15.07
B	< 1 mm	554	3.42
	~ 1 cm		3.24
C	< 1 mm	501	15.67
	~ 1 cm		14.99
D	< 1 mm	500	39.18
	~ 1 cm		37.50

To highlight this conclusion, the following text (Page 12, Line 292) has been revised and new figure (Supplementary Figure 21) has been added to the supplementary materials of the manuscript:

Original text: “The optical waveguiding capability of single crystals of **A**, **B**, and **C** was recorded, and their respective optical loss coefficients were found to be 0.41, 0.20, and 0.29 dB mm⁻¹ (Supplementary Figure 20).”

Revised text: “The optical waveguiding capability of single crystals of **A**, **B**, and **C** was characterized for crystals of lengths of 5 mm and 1.5 – 3.0 cm. The corresponding optical loss coefficients were found to be 0.41, 0.20, and 0.29 dB mm⁻¹ for the 5 mm length crystal (Supplementary Figure 20). When the crystal length is of centimeter scale, a small increase in the optical loss coefficient was observed (Supplementary Figure 21), indicating that the long crystals used for weaving retain commendable optical transmission properties.”

New Supplementary Figure 21. Optical waveguiding properties of long size crystals **A–C**. (a, c, e) Fluorescence spectra collected at one tip of the crystals of **A** (a), **B** (c) and **C** (e) with different distances between the tip and the excitation site of the laser. (b, d, f) The I_{tip}/I_{body} decays of crystals **A** (b), **B** (d) and **C** (f). The optical loss coefficients (α) were determined by single-exponential fitting of the function $I_{tip}/I_{body} = A\exp(-\alpha D)$, where I_{tip} and I_{body} are the fluorescence intensities of the outcoupled and incidence light respectively, A is the optical loss coefficient, and D is the distance between the excited site and the tip of crystals for collecting emission.

Comment: 4. The authors made the woven organic crystals into optically transmissive webs. Its luminescence mechanism is photoluminescence. When tightly woven organic crystal wires are stimulated point-to-point, adjacent parts of organic crystal wires are also stimulated to emit light, as shown in Figure 7h and 7i. How to avoid interference from other signals?

Response: We thank the Reviewer for the astute observation and for bringing up this point. To avoid the excitation of adjacent crystals when woven organic crystal wires are excited in a point-by-point manner, a piece of black paper was fixed to one side of the crystal tip where the signal as collected. In addition, in Figure 7, we just show images of multimode optically transmissive networks without relevant tests. Therefore, the above devices were not utilized.

To clarify this point, the following text (Page 12, Line 298) was revised and a new supplementary Scheme 1 was added to the Supporting Information:

Added text: “To avoid the excitation of adjacent crystals when woven organic crystal wires are stimulated point-to-point, a black paper mask with a pinhole of the size of the laser beam was applied to cover the crystals (Supplementary Scheme 1).”

New Supplementary Scheme 1. Schematic diagram of the optical waveguide testing setup used to study the optical transmission of the crystalline patches. The blue dashed line indicates the laser transmission direction, and the yellow dashed line indicates the direction of the light signal transmission of the excited crystals.

Comment: 5. *Hand weaving also has limitations. Crystal wires that are too short are not conducive to weaving, and long crystal wires may encounter challenges in luminescence efficiency and light transmission effects due to numerous defects during crystal growth. The authors should consider how to resolve this dilemma.*

Response: We thank the Reviewer for the suggestion and we do share their concerns. We note that this is the first report of the method, and we have used centimeter-sized crystals for weaving. However, we believe that it is possible to expand the concept to automate it further, where shorter crystals can be used with minimal human intervention in the process, similar to what is usually done for textiles. We envisage that processes for micromanipulation of small objects that are commonly available could be used for that purpose. For example, Chandrasekar et al. have demonstrated visualization and controlling of location by mechanical movements of tiny flexible crystals by a confocal microscope integrated into an atomic force microscope (*Adv. Opt. Mater.* 2020, 8, 2000959). This or similar methods could be applied in the future; for example, a similar setup could be used to weave smaller crystals, so that micro-crystalline patches could be obtained. As per our responses to the comments 2 and 3 above, crystals with a few defects and/or better quality can be selected by an automated screening process, and as we have shown, the luminescence efficiency and light transmission performance of these long crystals are comparable to that of short crystals.

Response to the comments from Reviewer #3:

Overall Comment: *I have carefully reviewed the manuscript “Woven organic crystals” and find the work to be a promising step toward harnessing the benefits of flexibility in molecular crystals for advanced applications. The manuscript presents a well-written study with a proposed strategy to address the identified challenges. However, to further enhance the acceptability of your study for practical applications, I recommend considering the following points:*

Comment: 1. *As this work serves as a prototype, it is crucial to outline your intentions for future advancements. Clearly define how you plan to refine and expand your study to make it more applicable to real-world scenarios and industrial applications.*

Response: We thank the Reviewer for the clear suggestions for improvements, especially within the scope of wider applications of the method that we propose. We do concur with the Reviewer that this work and the method that we propose provides a way to extend elongated single crystals into flexible, integrated 2D planar structures, with potential future applications in flexible electronics. In that regards, in the future, we plan to improve and expand this research in several ways: (1) Optimize the crystal growth method in order to obtain a large number of crystals of more uniform size and high quality, perhaps by using automated methods for controlling the crystal growth and screening crystal quality. This could be achieved, for example, by controlling the ambient humidity and temperature, and utilizing the solvent slow diffusion method to grow the crystals. (2) Optimize the weaving technique by applying a small probe or a cantilever with a microscope to be able to weave the crystals by automated micromanipulation. (3) We further envision exploring logical light/circuits and actuating intelligent soft robots based on woven crystal networks to make them more suitable and bring them closer to practical applications. We hope that the Reviewer will appreciate that the current work is the first, albeit necessary step towards those developments in the future.

Comment: 2. *In most assemblies, the porosity and thickness of crystals tend to be arbitrary, unlike the well-controlled molecular weaving observed in crystals. Investigate how these random distributions affect both the mechanical and luminescence properties of the crystals. Specifically, explore how increasing crystal thickness influences the tendency for elastic crystals to undergo plastic deformation under higher stress.*

Response: We thank the Reviewer for these suggestions, which are well received. We note that the arbitrary woven structure does not affect the original optical properties of the crystal. On the other hand, we agree that the effect of crystal thickness on the tendency of elastic crystals to deform under higher stresses warrants investigation, because it is relevant to the process of weaving as well as to the mechanical properties of the woven net.

Accordingly, we have approached three-point bending experiments in which four crystalline patches (3 × 3) of different thicknesses were tested. As shown in the new Supplementary Figure 11, these three-point bending tests were performed on four crystalline patches of **C** having different thicknesses and sizes. The average thicknesses of the crystals of compound **C** were about 32, 64, 124, 188 μm, respectively. Our experiments showed that with the increase in crystal thickness, the patch is capable to bear a larger load, while its ability to deform is decreased.

The following text and new Supplementary Figure 11 have been added to the revised manuscript (Page 8, Line 216) and the Supporting Information:

Added text: “To test the effect of crystal thickness on the deformation ability of the woven crystalline patches, four crystalline patches of compound **C** of different thicknesses and sizes were prepared. The average thicknesses of the crystals of **C** in the four patches were about 32, 64, 124, and 188 μm. Three-point bending tests were performed on the four crystalline patches (Supplementary Figures 11a, b). It was concluded that with the increase in crystal thickness, the patch can sustain a larger load, while its ability to be deformed is decreased

(Supplementary Figures 11c)."

New Supplementary Figure 11. (a, b) Fluorescence images of crystalline patches of compound **C** made of crystals with different thicknesses. The average thickness of the crystals in the patches from left to right is 32, 64, 124, and 188 μm. The patches are shown before (a) and after (b) the three-point bending experiments. (c) Load-displacement curves obtained from the crystalline patches of **C** shown in panels a and b.

Comment: 3. On page 5, line 121, you mention that "the crystal density is inversely correlated with the crystal thickness." Please provide a clear explanation and supporting evidence for this statement to validate its accuracy.

Response: We thank the Reviewer for the very careful reading. We acknowledge that the wording in this phrase is an unintentional error; the phrase "crystal density" should be "linear density", that is, it should refer to the average densities of crystalline yarns. As shown in Supplementary Figures 5b and 6a, the linear density is inversely correlated with the crystal thickness. We thank the Reviewer for pointing this out, and we confirm that we have carefully checked the remainder of the manuscript to make sure it is correct.

To correct this, the following text (Page 5, Line 140) has been revised in the manuscript:

Original text: "while the crystal density is inversely correlated with the crystal thickness (Supplementary Figures 6 and 7)."

Revised text: "while the linear density is inversely correlated with the crystal thickness (Supplementary Figures 6 and 7)."

Comment: 4. The treatment of the crystalline fabric with boiling water to remove the glue from the weft of the porous structures is an important step. However, assess the potential detrimental effects of this treatment on crystallinity. High temperatures during this process can lead to

thermal agitation of molecules, potentially causing disorder in the crystals.

Response: We agree with the Reviewer that the thermal treatment might lead to thermal agitation and cause disorder. To assess the potential effect of this treatment to the crystallinity, single crystal X-ray diffraction analysis was performed on crystal of **A** before and after immersion in boiling water. We found that the two single crystal structures and cell parameters are basically the same (new Supplementary Table 2). The results showed that the change in crystal quality is not detectable with X-ray diffraction methods, and based on this result, we conclude that the crystal retains an acceptable quality.

The following text and new Supplementary Table 2 have been added to the revised manuscript (Page 3, Line 85) and supporting information:

Added text: “In order to assess the potential detrimental effects of this treatment to crystallinity, single crystal X-ray diffraction analysis was performed on crystal of **C** before and after immersion in boiling water (for details, see the Supporting Information). The diffraction data confirmed that any effect of the treatment with boiling water was undetectable (Supplementary Table 2).”

Added text: “**X-ray crystallographic analysis.** Diffraction data of crystals of **C** before and after immersion in boiling water were collected on a Bruker D8 Venture diffractometer. The structures were solved with direct methods using the Olex2 suite of programs and refined by using the full-matrix least-squares on F^2 . The non-hydrogen atoms were refined anisotropically, and the positions of hydrogen atoms were calculated and refined isotropically.”

New Supplementary Table 2. Crystal data and structural refinement for the crystals **C** before and after immersion in boiling water

Formula	C ₁₄ H ₈ Br ₂ (original)	C ₁₄ H ₈ Br ₂ (after treatment with boiling water)
Formula weight	336.02	336.02
Temperature / K	273.15	273.15
Crystal system	triclinic	triclinic
Space group	$P\bar{1}$	$P\bar{1}$
$a / \text{\AA}$	4.0689(3)	4.0686(3)
$b / \text{\AA}$	8.9080(7)	8.9081(7)
$c / \text{\AA}$	16.1505(12)	16.1506(12)
$\alpha / ^\circ$	99.082(3)	99.099(3)
$\beta / ^\circ$	96.916(3)	96.916(3)
$\gamma / ^\circ$	100.402(3)	100.416(3)
Volume / \AA^3	561.83(7)	561.73(8)
Z	2	2
$\rho_{\text{calc.}} / \text{g cm}^{-3}$	1.986	1.987
μ / mm^{-1}	7.179	7.180
F(000)	324.0	324.0
Size / mm^3	0.20 × 0.20 × 0.20	0.20 × 0.20 × 0.20

Radiation type	MoK α ($\lambda = 0.71073$)	MoK α ($\lambda = 0.71073$)
$\theta_{min} / ^\circ$	5.17	5.17
$\theta_{max} / ^\circ$	55.092	55.112
Index ranges	$-5 \leq h \leq 5, -11 \leq k \leq 11, -20 \leq l \leq 20$	$-5 \leq h \leq 5, -11 \leq k \leq 11, -20 \leq l \leq 20$
Reflections collected	2248 wR_4	24193
Independent reflections	2596 [$R_{int} = 0.0329, R_{sigma} = 0.0158$]	2596 [$R_{int} = 0.0316, R_{sigma} = 0.0147$]
Data/restraints/parameters	2596 / 0 / 145	2599 / 0 / 145
Goodness-of-fit on F^2	1.224	1.189
Final R indexes [$I \geq 2\sigma(I)$]	$R_1 = 0.0486, wR_2 = 0.1210$	$R_1 = 0.0400, wR_2 = 0.1029$
Final R indexes [all data]	$R_1 = 0.0539, wR_2 = 0.1262$	$R_1 = 0.0446, wR_2 = 0.1068$
Largest diff. peak / hole / e \AA^{-3}	0.59 / -1.97	0.51 / -1.46

Comment: 5. *Given that the assemblies consist of different crystals with varying degrees of elastic flexibility and distinct melting temperatures, it becomes crucial to establish correlations between their mechanical properties after the treatment with boiling water. Provide the homologous temperature of the crystal system to better illustrate the possible thermal effect of this treatment.*

Response: We thank the Reviewer for this suggestion, which made us think about the effect of water treatment. In order to address this comment, three-point bending and tensile tests of crystal of **C** before and after immersion in boiling water were conducted. However, as shown in Figure 1c, we note that only the leftmost part of the weft crystals of crystalline patch was treated with boiling water. This, at least in principle, should not affect the mechanical properties of the intermediate (woven) section of the crystalline patch. In order to explore the effect of boiling water treatment on the mechanical properties of the crystal, we carried out three point bending and tensile tests of crystal of **C** before and after soaking in boiling water (new Supplementary Figure 13). We found that the bending modulus and tensile modulus of the crystals **C** after boiling water treatment were basically identical to those of the untreated crystals of **C**. The results show that this treatment has small or no effects on the mechanical properties of the crystals. In addition, we mentioned that boiling water treatment has basically no effect on the quality of single crystals. We note that crystals of **A–C** have good thermal stability (melting points: 205, 197, and 227 °C; Supplementary Figure 11), and they do not undergo phase transitions around 100 °C, so we conclude that this treatment has almost no thermal effect on the crystals **A–C**.

In order to clarify these points, the following text and new Supplementary Figure 13 have been added to the revised manuscript (Page 9, Line 226) and the Supporting Information:

Added text: “Furthermore, to explore the effect of treatment with boiling water on the mechanical properties of the crystals, we carried out three point bending and tensile tests on crystals of **C** before and after soaking in boiling water (new Supplementary Figure 13). The bending modulus before and after the treatment were 4.56 – 4.81 GPa and 4.31 – 4.35 GPa, and the tensile modulus before and after treatment were 1.69 – 2.05 GPa and 1.57 – 1.88 GPa, respectively. The results indicate that the crystals are thermally stable and the treatment does not have a significant effect on their mechanical properties.”

New Supplementary Figure 13. (a, b) Load–displacement (a) and stress-strain (b) profiles of crystals of **C** before and after immersion in boiling water obtained from three-point bending tests. (c, d) Load–displacement (a) and stress-strain (b) profiles of crystals of **C** before and after immersion in boiling water obtained from tensile tests.

Comment: 6. *Apart from plain weaving, incorporating twisted crystals into the fabric introduces two-dimensional plasticity in addition to elastic properties. Investigate how such crystals contribute to the fabric's overall plastic nature, particularly under higher stress conditions.*

Response: Following this suggestion, and in order to study how the torsional crystal affects the overall plastic properties of the woven crystalline patches, three-point bending test was carried out on four patches consisting of crystals of **D** in their straight and twisted states. The results are summarized in the new Supplementary Figure 12.

The following text and new Supplementary Figure 12 have been added to the revised manuscript (Page 9, Line 222) and Supporting Information:

Added text: “In addition, three-point bending tests were also performed on twisted crystals of **D** to study the contribution of these crystals to the fabric's overall plastic nature. As shown in Supplementary Figure 12, compared to woven patches consisting of straight crystals of **D**, the patches made of twisted crystals of **D** are more susceptible to deformation. Twisted patches are more likely to develop shearing forces (stress on both two face of the crystal), and are more prone to fragmentation.”

new Supplementary Figure 12. (a, b) Fluorescence images of four crystalline patches of **D** in straight and twisted states before (a) and after (b) three-point bending experiments. (c) Load-displacement curves obtained from the crystalline patches of **D**.

Comment: *In conclusion, the manuscript shows significant promise in exploring the potential of flexible woven organic crystals for various applications. Addressing the above-mentioned comments will enhance the completeness and applicability of the study. I recommend revising the manuscript in light of these suggestions, and I look forward to seeing your work published after these improvements.*

Response: We thank the Reviewer once again for the very careful reading and all the thoughtful comments that initiated some thinking, and ultimately have provided material to improve the claims in our manuscript. We hope that in the revised version we have succeeded in addressing the Reviewer's comments to their satisfaction.

REVIEWER COMMENTS

Reviewer #1 (Remarks to the Author):

In my opinion, authors have addressed all the comments by both the referees satisfactorily. I am fully satisfied to recommend the article for publication in its current form.

Reviewer #2 (Remarks to the Author):

In this revised version, the authors elaborated on the materials quality issues (i.e. defects) of organic crystal wires and supplemented the morphological and optical characterization tests of the wires. However, there are still some concerns that need to be addressed.

1. The authors only tested the optical loss of wires A, B, and C separately, rather than the optical loss of the woven wires with interconnected network. It is curious to know how the optical loss of the woven wires, for example, a mixture of wires A and B or B and C, etc. To demonstrate the practicality of this weaving technique, it is suggested to add relevant characterizations.

2. Another issue, the authors emphasize that the process of preparation of the crystals is out of the scope of their proposed weaving method, but actually their proposed weaving approach is self-limited by the reproducibility and yield of the organic crystal wires. Just as the authors said, thousands of long needle-like crystals were obtained by using controllable crystal growth methods in this work, but the truth is the length and width of the prepared crystal line are too random, especially for the width. One can see there is a considerable difference of the individual wire width even though they are prepared using the same preparation parameters (see in the new Supplementary Figure 2), which is unfavorable for the overall weaving device. In my opinion, this method is not universal and controllable. I suggest the authors to compare it with the ones fabricated with sophisticated electrospinning techniques, and at least, the statistics of the width (diameter) of the wires prepared from each condition should be provided.

Reviewer #3 (Remarks to the Author):

The authors have addressed all the comments and addressed the concerns outlined in my review report. The report not only remains engaging but also introduces a fresh perspective to the study of flexible crystals, as I had previously noted. Therefore, I recommend its acceptance in its current format for publication in Nature Communications.

Response to Reviewers' comments on the manuscript "Woven Organic Crystals" by Lan and the collaborators

The reviewers' comments are written in *black italic font*.

Our responses to the comments are written in **blue regular font**.

Changes made to the main text are written in **red regular font**.

Response to the comments from Reviewer #2:

Comment: 1. *The authors only tested the optical loss of wires A, B, and C separately, rather than the optical loss of the woven wires with interconnected network. It is curious to know how the optical loss of the woven wires, for example, a mixture of wires A and B or B and C, etc. To demonstrate the practicality of this weaving technique, it is suggested to add relevant characterizations.*

Response: We thank the Reviewer for the suggestion. Indeed, it would be important to study the optical loss of the woven organic crystalline wires. Accordingly, the optical waveguide properties were measured for the woven wires of the patch **A#B#C** and in the revised version the results were added as new Supplementary Figure 22. According to the previous testing method, the optical loss coefficients (OLCs) of the warp and weft crystals **A**, **B** and **C** of the patch **A#B#C** were measured. One crystal was tested at a time, and a total of three weft crystals and three warp crystals were measured. The results showed that all OLC values of the crystals are close to the OLC values of crystals **A**, **B** and **C**, respectively.

To clarify this point, the following text (Page 12, Line 302) and new Supplementary Figure 22 have been added in the revised version of the Supporting Information:

Added text: "Furthermore, to evaluate the effect of weaving on the optical transmission performance of crystals, the same tests were performed on six crystals (the warp and weft crystals **A**, **B** and **C**) of the patch **A#B#C**. The OLC values of the woven crystals are close to the aforementioned values for crystals **A**, **B** and **C** (Supplementary Figure 22)."

New Supplementary Figure 22. Optical waveguiding properties of weft and warp crystals **A–C** of the patch **A#B#C**. (a, c, e) Fluorescence spectra collected at one tip of the weft and warp crystals **C** (a), **B** (c) and **A** (e) with different distances between the tip and the excitation site of the laser. (b, d, f) The $I_{\text{tip}}/I_{\text{body}}$ decays of the weft and warp crystals **C** (b), **B** (d) and **A** (f). The optical loss coefficients (α) were determined by single-exponential fitting of the function $I_{\text{tip}}/I_{\text{body}} = A \exp(-\alpha D)$, in which I_{tip} and I_{body} are the fluorescence intensities of outcoupled and incidence light respectively, A is the optical loss coefficient, and D is the distance between the excited site and the tip of crystals for collecting emission.

Comment: 2. Another issue, the authors emphasize that the process of preparation of the crystals is out of the scope of their proposed weaving method, but actually their proposed weaving approach is self-limited by the reproducibility and yield of the organic crystal wires. Just as the authors said, thousands of long needle-like crystals were obtained by using controllable crystal growth methods in this work, but the truth is the length and width of the prepared crystal line are too random, especially for the width. One can see there is a considerable difference of the individual wire width even though they are prepared using the same preparation parameters (see in the new Supplementary Figure 2), which is unfavorable for the overall weaving device. In my opinion, this method is not universal and controllable. I suggest the authors to compare it with the ones fabricated with sophisticated electrospinning techniques, and at least, the statistics of the width (diameter) of the wires prepared from each condition should be provided.

Response: We thank the Reviewer for their suggestion. We agree with the Reviewer in that the length and width of the prepared crystal line are fairly random, especially with regards to the width. As it has been very well established in the crystal growth science, the aspect ratio of the crystals depends on the conditions of crystallization, and, except for some very special cases, is beyond the control of the experimenter. We would like to note that this part of the process is not part of our work, which focuses on the weaving and the properties of the woven patches, given that crystals of high aspect ratio (in general) are available. In order to respond to the Reviewer’s comment, we tried to quantify this variation in the patches that were prepared by measuring the dimensions of the warp and weft crystals for 11 crystalline patches, and the results are shown in Supplementary Tables 3 and 4. As shown in the new Supplementary Table 5, we further evaluated the average width of the crystals in these patches together with the respective standard deviation. In addition, the dimensions of the reported polymer fibers using other techniques, such as electrospinning, are also shown in the table for comparison. We hope that these statistics provide at least an insight into the expected spread of sizes that are applicable (as proven by the woven patches) in the case of molecular crystals.

Accordingly, the following text has been added (Page 5, Line 135) in the revised manuscript, and new Supplementary Table 5 has been added in the revised version of the Supporting Information:

Added text: “Furthermore, the average width and standard deviation of the crystals in the patches were assessed and compared to the dimensions of reported fibrous materials that are used for weaving (Supplementary Table 5).”

New Supplementary Tables 5. Comparison of the dimensions of the woven crystals in crystalline patches and the woven structures obtained by other methods such as electrospinning

Woven structures	Dimensions (width or diameter) / µm	Standard deviation / µm
A1	104.26	16.28
A2	195.05	29.07
B	75.79	19.44
C1	102.43	21.70
C2	263.36	46.69
C3	140.88	15.68
A#B	103.12	17.73
A#C	188.31	24.79
B#C	72.33	23.80
A#B#C	85.52	26.62
D (twisted)	67.94	16.60
PVP (doped-crystals) ^[5]	2	—
PVDF fibers ^[6]	40–120	—
PAN nanofibers ^[7]	0.28–0.84	—
Viscose composite yarns ^[7]	11.57	—
PAN fibers ^[8]	0.35–0.50	—

GeIMA fibers ^[9]	2.18	0.52
Crosslinked gelatin fibers ^[9]	1.76	0.45
PCL NY ^[10]	0.4533	0.0553
PCL/SF-4/1 NY ^[10]	0.6133	0.0488
PCL/SF-3/2 NY ^[10]	0.9508	0.0898
RP-PCL ^[11]	220	25
EHD-PCL ^[11]	205	61

REVIEWERS' COMMENTS

Reviewer #2 (Remarks to the Author):

In this revised version, I appreciated that the authors tried their best to address my remaining concerns by testing the optical loss of the woven wires and provided a statistic for the dimensions of their woven wires.

For a more clear understanding, I suggest the authors to provide an additional sketch showing the actual patch A#B#C and/or other combinations that they really used to test the optical waveguiding properties, and the set-up they used to test the fluorescence spectra showing the position of the tips and the excitation sites of the crystal wires. That would be more easier for the reader to understand their woven wires and the corresponding optical properties.

Response to Reviewers' comments on the manuscript "Woven Organic Crystals" by Lan and the collaborators

The review comments are written in *black italic font*

Our responses to the comments are written in **blue regular font**

Changes made to the main text are written in **red regular font**

Response to the comments from Reviewer #1:

Overall Comment: *In this revised version, I appreciated that the authors tried their best to address my remaining concerns by testing the optical loss of the woven wires and provided a statistic for the dimensions of their woven wires.*

For a more clear understanding, I suggest the authors to provide an additional sketch showing the actual patch A#B#C and/or other combinations that they really used to test the optical waveguiding properties, and the set-up they used to test the fluorescence spectra showing the position of the tips and the excitation sites of the crystal wires. That would be more easier for the reader to understand their woven wires and the corresponding optical properties.

Response: We thank the Reviewer again for the careful reading and all the comments, which we tried to address to the best of our ability. For a clearer understanding of optical waveguide test details, we have added new Supplementary Figure 29 in the revised manuscript (Page 8, Line 265) and supporting information (Page 19).

New Supplementary Figure 29. (a, b) The Schematic (a) and set-up (b) for a crystalline patch used to collect the fluorescence spectra. (c) Optical waveguides of patch **A#B#C** observed from side (left) and front directions (right). The blue arrow represents a 355 nm laser. The green dotted arrows represent the direction of receiving the optical signal. The dashed blue arrows represent the direction in which the micro control patches move.